# SALSA: Soup-based Alignment Learning for Stronger Adaptation in RLHF

## Abstract

In Large Language Model (LLM) development, Reinforcement Learning from Human Feedback (RLHF) is crucial for aligning models with human values and preferences. RLHF traditionally relies on the Kullback-Leibler (KL) divergence between the current policy and a frozen initial policy as a reference, which is added as a penalty in policy optimization algorithms like Proximal Policy Optimization (PPO). While this constraint prevents models from deviating too far from the initial checkpoint, it limits exploration of the reward landscape, reducing the model's ability to discover higher-quality solutions. As a result, policy optimization is often trapped in a narrow region of the parameter space, leading to suboptimal alignment and performance. This paper presents **SALSA** (**S**oup-based **A**lignment **L**earning for **S**tronger **A**daptation), a novel approach designed to overcome these limitations by creating a more flexible and better located reference model through weight-space averaging of two independent supervised fine-tuned (SFT) models. This model soup allows for larger deviation in KL divergence and exploring a promising region of the solution space without sacrificing stability. By leveraging this more robust reference model, SALSA fosters better exploration, achieving higher rewards and improving model robustness, out-of-distribution generalization, and performance. We validate the effectiveness of SALSA through extensive experiments on popular open models (Llama2-7B, Mistral-7B, and Gemma-2B) across various benchmarks (MT-Bench, Arena-Hard, UltraFeedback), where it consistently surpasses PPO by fostering deeper exploration and achieving superior alignment in LLMs.

## 1 Introduction

Large language models (LLMs) have revolutionized natural language processing (NLP) by demonstrating remarkable capabilities in understanding and generating human language. These models, powered by vast amounts of data and advanced neural architectures, have set new benchmarks in various NLP tasks, from machine translation to conversational agents. Despite these advancements, aligning LLMs with human values and preferences remains a significant challenge. Misalignment can lead to undesirable behaviors, including generating biased or inappropriate content, which undermines the reliability and safety of these models (Kenton et al., 2021; Bender et al., 2021; Bommasani et al., 2021; Gehman et al., 2020).

Reinforcement Learning from Human Feedback (RLHF) has become a promising technique for aligning large language models (LLMs) with human preferences. By fine-tuning LLMs based on human feedback, RLHF guides models towards more human-aligned behaviors, improving truthfulness, helpfulness, and harmlessness while maintaining the generation of high-probability, correct answers (Christiano et al., 2017). Reward-based RLHF methods utilize a reward model to determine the reward for a given (prompt, response) pair. The policy model is then optimized by maximizing the average reward through reinforcement learning algorithms like Proximal Policy Optimization (PPO) (Schulman et al., 2017). A crucial aspect of RLHF is the use of a reference model to compute the Kullback-Leibler (KL) divergence penalty, which prevents the fine-tuning process from deviating too far from the original model (Ziegler et al., 2019). This approach ensures that the policy remains close to the initial model, reducing the risk of generating nonsensical responses.

While effective, reliance on a single reference model can be limiting. The KL penalty term constrains the policy model to stay close to the initial supervised fine-tuning (SFT) model, restricting its ability to fully explore the solution space for higher-reward models. This constraint can lead to suboptimal alignment and a lack of robustness in the training process, increasing the risk of generating nonsensical outputs. Ensuring that the reference model is already positioned in a robust space can mitigate this issue, allowing for more confident exploration without compromising output quality.

To address this limitation, we propose SALSA. It integrates a *"model soup"* as the reference model within the RLHF framework. A model soup is constructed by performing weight-space averaging of multiple independently supervised fine-tuned models that demonstrate comparable performance. This method leverages the principle that fine-tuned models from the same pre-trained initialization often reside in a shared low-error basin in the loss landscape, enabling effective weight interpolation without compromising accuracy. As evidenced by Wortsman et al. (2022), this approach results in significant improvements in both in-distribution and out-of-distribution generalization. The key advantage of *model soup* lies in its ability to harness the complementary strengths of diverse models, reducing variance and improving robustness, while maintaining computational efficiency. This makes the *"model soup"* reference model a superior choice for improving the stability and reliability of the RLHF training process compared to relying on a single reference model.

In this paper, we demonstrate the effectiveness of SALSA through comprehensive experiments. We apply SALSA to **Llama2-7B**, **Mistral-7B**, and **Gemma-2B** and benchmark the results against standard evaluation datasets, including MT-Bench, Arena-Hard, and UltraFeedback—the latter being used for RLHF training in our experiments. Our findings reveal that weight space averaging is a straightforward yet effective approach for aligning LLMs with human preferences, and enhancing their performance on real-world-like datasets.

In particular our contributions are following:

- We demonstrate that the reward in the region near the model soup is inherently superior to that of the original SFT model. The improvement in reward is a newly observed phenomenon and is complementary to the improvement in accuracy of model soups. We further show having model soups as reference point of RLHF results in higher reward outcomes. (Section 4.2).

- Drawing from these observations, we propose SALSA, a novel approach for implementing RLHF that utilizes the model soup as the reference model.

- We perform a comprehensive evaluation across diverse benchmarks and models, demonstrating that SALSA consistently outperforms PPO (Section 4.3).

The remainder of this paper is organized as follows: Section 2 reviews related work on RLHF and model averaging techniques. Section 3 details our methodology for creating and integrating the model soup in the RLHF process. Sections 4.1 through 4.4 present the experimental results and analysis of model averaging for better alignment.

## 2 RELATED WORKS

### 2.1 MODEL SOUP

Model soups build on the findings of (Neyshabur et al., 2020), which showed that independently fine-tuned models often lie within the same loss basin, allowing their weights to be successfully combined through interpolation. (Wortsman et al., 2022) extended this, demonstrating that averaging fine-tuned model weights can improve performance compared to using a single model. Unlike traditional ensembling, model soups require no extra memory or inference time, while still enhancing robustness. This method has shown consistent performance gains across NLP and image classification tasks (Wortsman et al., 2022; Izmailov et al., 2018).

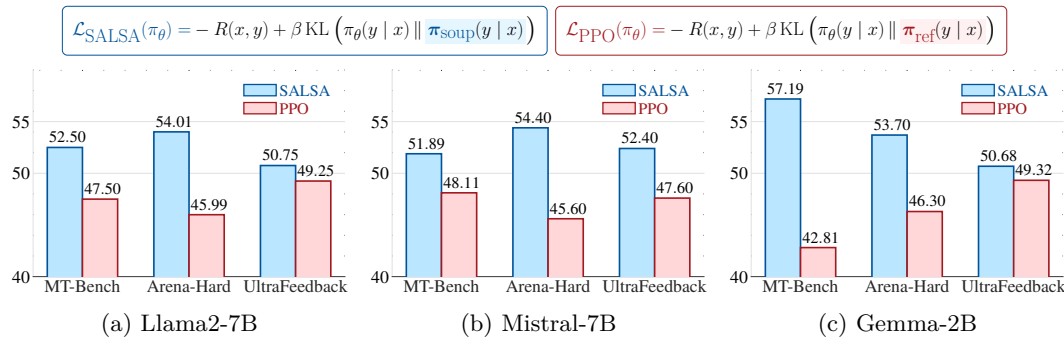

Figure 1: Comparison of SALSA and PPO. The main difference between SALSA and PPO is in the reference model within KL divergence of loss. SALSA consistently outperforms PPO across different models and tasks.

## 2.2 REINFORCEMENT LEARNING FROM HUMAN FEEDBACK (RLHF)

Large language models (LLMs) have achieved significant success across tasks, thanks to fine-tuning methods like Supervised Fine-Tuning (SFT) and Reinforcement Learning from Human Feedback (RLHF) (Ouyang et al., 2022; Touvron et al., 2023a; Bai et al., 2022; Anil et al., 2023). RLHF has been crucial for aligning models with human preferences, enabling outputs that are more helpful and aligned with societal values (Ziegler et al., 2019; Stiennon et al., 2020; Achiam et al., 2023).

RLHF methods are divided into reward-based and reward-free approaches. Reward-based methods rely on training a reward model to guide policy optimization, typically with Proximal Policy Optimization (PPO) (Schulman et al., 2017; Gao et al., 2023). PPO is widely used for its balance of stability and exploration (Ouyang et al., 2022). Studies have also examined the importance of hyperparameter tuning and reward model quality to avoid overoptimization (Casper et al., 2023; Zheng et al., 2023a). Reward-free methods, such as Direct Preference Optimization (DPO), bypass the reward model by directly optimizing human preference data (Rafailov et al., 2023; Liu et al., 2023; Yuan et al., 2023). While DPO simplifies training and performs well on tasks like instruction following (Touvron et al., 2023a), it struggles with out-of-distribution data, and its generalization capabilities are limited (Yuan et al., 2024; Xu et al., 2024).

In this paper, we focus on Proximal Policy Optimization (PPO) for its demonstrated robustness in managing large models and diverse tasks. Studies such as (Xu et al., 2024) highlight the limitations of Direct Preference Optimization (DPO), which tends to find biased solutions when confronted with out-of-distribution responses and is highly sensitive to distribution shifts. In contrast, PPO effectively mitigates these challenges through the use of reward models and KL divergence regularization (Ouyang et al., 2022; Schulman et al., 2017). This enables PPO to consistently outperform DPO in tasks like dialogue and code generation, proving its reliability across alignment challenges (Xu et al., 2024).

Recent RLHF research has introduced novel approaches to improve upon traditional methods, particularly in addressing the limitations of using reference models. SimPO (Meng et al., 2024) eliminates the need for a reference model by optimizing the average log probability of sequences with a reward-free approach. This reduces computational costs and improves memory efficiency over traditional DPO, achieving better performance across benchmarks like AlpacaEval and Arena-Hard. SimPO underscores the limitations of static reference models in DPO and advocates for scalable, reference-free approaches. Conversely, (Gorbatovski et al., 2024) introduces a dynamic reference model that evolves throughout training using Trust Region methods. This dynamic approach allows the reference model to adapt alongside the policy, preventing constraints imposed by outdated checkpoints and enabling more effective generalization and alignment with human preferences.

The aforementioned papers have identified that the reference model may limit the optimization process in DPO, and they propose innovative solutions to address this issue. Similarly, we

aim to solve this problem within the PPO framework. We adopt a model soup approach, which averages the weights of fine-tuned models. This method allows the optimization process in PPO to explore a broader solution space, offering greater flexibility and enhancing alignment performance.

## 3 Method

In this section we present our method, SALSA (Soup-based Alignment Learning for Stronger Adaptation). We begin with a brief overview of Proximal Policy Optimization (PPO), and model soup, followed by a detailed description of our approach.

### 3.1 RLHF

In our main experiments, we focus on reward-based RLHF, in particular Proximal Policy Optimization (PPO). The conventional framework for reward-based RLHF consists of several key stages as follows.

**Supervised Fine-Tuning (SFT).** The initial stage of alignment involves supervised fine-tuning, where a pre-trained language model is refined using a high-quality instruction dataset.

**Reward Model.** Reward-based RLHF involves training a reward model, which is typically initialized from the SFT model. In this process, the logit layer of the SFT model is replaced by a new linear layer. This linear layer takes the embedding of the last token and outputs a scalar reward. Higher rewards indicate better samples. For a given prompt $x$ and a pair of responses $y_w$ (chosen) and $y_l$ (rejected), the loss function is optimized as:

$$\mathcal{L}(R_\theta) = -\log \sigma(R_\theta(x, y_w) - R_\theta(x, y_l)), \tag{1}$$

where $R_\theta(\cdot, \cdot)$ is the reward model, $\theta$ denotes its parameters, $\sigma$ indicates the sigmoid function.

**Policy Training.** The last phase of RLHF is dedicated to training the policy model, which is initialized from a reference model, typically the SFT model. Based on a recent study (Xu et al., 2024), PPO performs better in distribution shifts and results in superior alignment with human preferences across challenging tasks like code generation and dialogue systems. Therefore, we selected PPO as the training algorithm. The goal is to optimize the policy model to maximize the reward for a given prompt $x$ and its generated response $y$, while also minimizing the KL divergence between the policy model and the reference model. The overall loss function for this stage is given by:

$$\mathcal{L}_{\mathcal{PPO}}(\pi_\theta) = -R(x, y) + \beta \text{KL}\left(\pi_\theta(y \mid x) \,\|\, \boldsymbol{\pi}_{\text{ref}}(y \mid x)\right) \tag{2}$$

where $\pi_\theta$ is the policy model, $\pi_{ref}$ is the reference model, and $R(., .)$ is the trained reward model.

### 3.2 Model Soup

The concept of a model soup, introduced in Wortsman et al. (2022), aims to enhance model performance by averaging the weights of multiple pre-trained networks. This technique combines the parameters of independently trained models to produce a more robust outcome, leveraging the strengths of each. There are three primary strategies for constructing a model soup: uniform, greedy, and learned. In our approach, we employ its most basic strategy, *i.e.*, the uniform method, where the parameters of separate SFTs are averaged. Investigating other mixing strategies for RLHF and comparing them is left as future work. Formally, we construct our soup model using two SFT models: $\pi_{ref}$, which serves as the initialization of the policy model in the PPO framework, and $\pi_{other}$, an additional SFT model trained on the same data with a different random seed. Their weights denoted as $\theta$ is averaged using a coefficient $\alpha$:

$$\theta_{soup} = (1 - \alpha)\theta_{ref} + \alpha\theta_{other} \tag{3}$$

In our experimental setup, as shown in Figure 4a, setting $\alpha$ to 0.5 produces the best results. This will be further discussed in the next section, where we also explain our proposed method, SALSA.

### 3.3 SALSA

The KL term in equation 2 ensures that the model remains closely aligned with the reference model. This term is essential because optimizing solely for the reward can result in the generation of nonsensical outputs. However, it also imposes a constraint by preventing the model from deviating significantly from the initial reference model. To address this limitation, we propose replacing the KL term in 2, with the following loss function:

$$\mathcal{L}_{\mathcal{SALSA}}(\pi_\theta) = -R(x, y) + \beta \text{KL} \left( \pi_\theta(y \mid x) \, \| \, \boldsymbol{\pi}_{\text{soup}}(y \mid x) \right) \tag{4}$$

As mentioned in Equation 3, $\pi_{soup}$ refers to a model soup, which is the result of averaging two independently trained supervised fine-tuned models (SFTs), including the reference model. Since the policy model $\pi_\theta$ is initialized from the reference model $\pi_{ref}$, substituting the KL term in equation 2 with the KL term in 4 which is the model soup of $\pi_{ref}$ and $\pi_{other}$ allows the policy model to search around averaged model, thereby enabling exploration of a broader and more promising parameter space. The primary distinction between the loss terms in equations 2 and 4 is that the soup model is used in place of the reference model. This substitution keeps our approach straightforward to implement while proving highly effective. Our experiments demonstrate improved performance, resulting in higher win rates over PPO across three models and three datasets. These consistent results indicate SALSA's effectiveness in various settings.

## 4 EXPERIMENTS

### 4.1 EXPERIMENTAL SETUPS

In this section, we outline our experimental setup. We use three models: Llama2-7B (Touvron et al., 2023b), Mistral-7B (Jiang et al., 2023), and Gemma-2B (Team et al., 2024). For Supervised Fine-Tuning (SFT), we employ the UltraChat-200k dataset (Ding et al., 2023). The UltraFeedback dataset (Cui et al., 2023) is utilized for both training the reward model and optimizing preferences. All experiments across the three stages (SFT, reward model, RLHF) are conducted using the TRL library by HuggingFace (von Werra et al., 2020). For both PPO and SALSA, we use the KL coefficient $\beta$ that achieves the highest win rate. Further details on the hyperparameter settings are available in Appendix A.

**Evaluation Benchmarks.** We evaluate the effectiveness of our method using three well-established instruction-following benchmarks: MT-Bench (Zheng et al., 2024), Arena-Hard v0.1 (Li et al., 2024), and UltraFeedback (Cui et al., 2023) test dataset. MT-Bench comprises 80 questions across 8 categories, while Arena-Hard is an enhanced version of MT-Bench, featuring 500 well-defined technical problem-solving questions.

In our evaluation process, we used the datasets described in the previous section to generate samples. Pairwise comparisons were then conducted using GPT-4-Turbo as the judge model, following the "LLM-as-a-judge" methodology. The prompt used for our judge model is provided in Figure 7 in the Appendix. For evaluation, we utilized the FastChat repository (Zheng et al., 2023b). For the MT-Bench questions, which involve two rounds, we generated and evaluated outputs for each round.

### 4.2 REWARD ANALYSIS

We hypothesize that $\pi_{soup}$ resides in a region of the parameter space associated with generally higher rewards, suggesting that models explored in this vicinity could generate responses with increased reward values, in addition to the improved loss observed in the original model soup paper Wortsman et al. (2022). To test this hypothesis, we conducted an analysis along the interpolation line between two SFT models, $\pi_{ref}$ and $\pi_{other}$, for Gemma-7B and Llama2-7B, examining how the mean reward changes for the MT-Bench dataset. These rewards were calculated on raw models prior to RLHF process. As illustrated in Figure 2a and 2b, we observe that the mean reward on the dataset increases as we move towards the midpoint between these two models, and subsequently decreases beyond this point. This

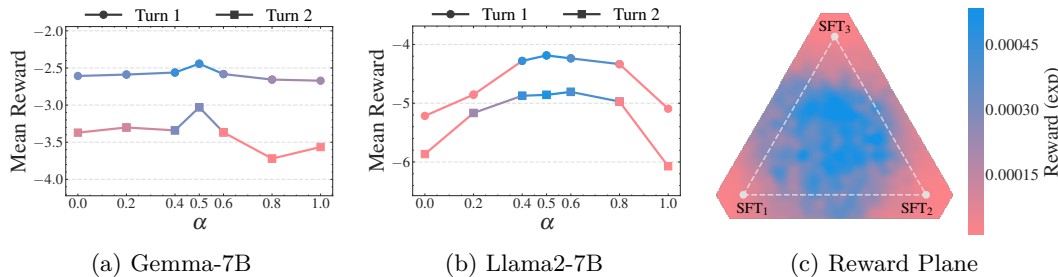

(a) Gemma-7B  (b) Llama2-7B  (c) Reward Plane

Figure 2: (a) The reward of model averaging for Gemma-7B peaks in $\alpha = 0.5$. (b) The same phenomenon is seen for Llama2-7b (c) The heatmap of rewards of Llama2-7B around 3 SFT model in a Barycentric space. Inside the triangle which is closer to average of 3 models has significantly higher reward than outside triangle. This shows model soups are in a more promising region for searching.

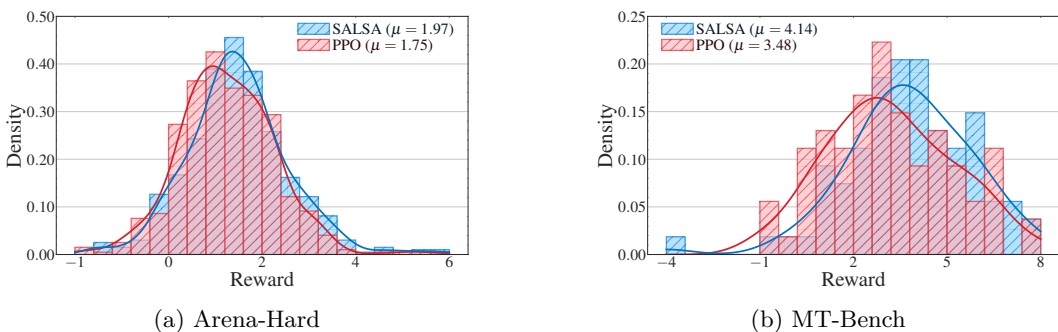

(a) Arena-Hard  (b) MT-Bench

Figure 3: Comparison of reward distributions between SALSA and PPO for the Llama2-7B model. SALSA gets higher reward in average across both datasets.

pattern indicates that $\pi_{soup}$ - which represents the average of the SFTs - resides in a region of the parameter space associated with higher rewards further supporting our hypothesis.

Next, based on our observations of improved rewards from model soup combining two SFT models, we extended our experiments to explore reward behavior within the space defined by three SFT models. These models were trained on UltraChat-200k using a pretrained Llama2-7B, each initialized with different random seeds. Specifically, we evaluated rewards on a plane defined by these three models, both inside and outside this space. Figure 2c presents the results of these experiments. The vertices of the dotted-line triangle represent the three SFT models. As shown, moving towards the midpoint between any two SFT models consistently leads to increased rewards. A similar trend is observed as we approach the center of the triangle. Additionally, the rewards for points outside the triangle decreases. Since PPO tends to find solutions near the vertices of the triangle (due to its reliance on the initial SFT model), the corresponding rewards in these regions are lower, as shown in the figure. This suggests that PPO may struggle to guide the model towards areas associated with higher rewards, which are located more centrally between the SFT models

Third, we hypothesize that SALSA's ability to explore a better parameter space, enables the model to discover regions with higher rewards while maintaining the generation of sensible responses. This expanded exploration ultimately results in SALSA's superior performance over PPO. To validate this hypothesis, we compared the reward distributions for responses generated by SALSA and PPO across the MT-Bench and Arena-Hard datasets. Figure 3 illustrates the reward distributions for both methods. The plot reveals that the reward distribution for SALSA is shifted towards higher values compared to PPO. This rightward shift is consistent across both datasets, indicating that SALSA consistently generates responses associated with higher rewards. Furthermore, the mean reward for SALSA is higher than that of PPO in both datasets, further supporting our hypothesis.

Table 1: Comparison of Adjusted Win Rates across Models and Datasets

| Dataset | Model | SALSA vs PPO | SALSA vs SFT |
|---------|-------|:------------:|:------------:|
| **MT-Bench** | Llama2-7B | **52.50** | **52.50** |
| | Mistral-7B | **51.89** | **55.94** |
| | Gemma-2B | **57.19** | **56.88** |
| **Arena-Hard** | Llama2-7B | **54.01** | **54.70** |
| | Mistral-7B | **54.40** | **55.70** |
| | Gemma-2B | **53.7** | **53.8** |
| **UltraFeedback** | Llama2-7B | **50.75** | **52.23** |
| | Mistral-7B | **52.40** | **51.93** |
| | Gemma-2B | **50.68** | **54.32** |

Furthermore our findings indicate that employing a model soup as the reference model permits greater deviation in KL divergence. This increased flexibility allows the policy to investigate a more extensive area within the solution space. More info in this regard can be found in Appendix A.4.

### 4.3 MAIN RESULTS

Figure 1 visualizes the win rate comparison between SALSA and PPO, where SALSA achieves notable win rates of 54.01% for the Llama2-7B model and 54.40% for the Mistral-7B model on the challenging Arena-Hard dataset. Table 1 provides a comparative analysis of SALSA against the original PPO and SFT models for Llama2-7B, Mistral-7B, and Gemma-2B. The results indicate that SALSA consistently outperforms both PPO and SFT, demonstrating its effectiveness. Based on the detailed results in tables 2, 3, and 4 in the Appendix results, PPO and SFT often perform similarly on MT-Bench and Arena-Hard, likely because UltraFeedback and UltraChat, which are utilized for SFT, Reward Modeling, and RLHF, are considered out-of-distribution for these benchmarks. As a result, a basic version of PPO does not significantly outperform SFT. However, SALSA's robustness to out-of-distribution data (derived from weight averaging and model soup techniques) delivers improvements of up to 57% and 54% on these datasets. These results underscore SALSA's effectiveness in enhancing out-of-distribution robustness in RLHF training while maintaining competitive performance for in-distribution, and emphasize SALSA's superior exploration capabilities and improved reward optimization, resulting in better task alignment and overall performance. Additional detailed results are available in tables 2, 3, and 4 in the Appendix. Also a qualitative comparison of responses generated by SALSA and PPO is presented in Figure 8 in Appendix.

### 4.4 ABLATION STUDY

We explored using alternative reference points along the line between $\pi_{\text{ref}}$ and $\pi_{\text{other}}$ by varying the $\alpha$ value in Equation 3. For each $\alpha$, we adjusted the KL divergence to achieve the optimal win rate over PPO. This systematic adjustment allowed us to observe how the win rate varies with $\alpha$, as shown in Figure 4a which reveals a clear trend: the adjusted win rate increases as $\alpha$ approaches 0.5, peaking at this midpoint before declining at higher values. Notably, using $\pi_{other}$ alone does not enhance performance, yielding an adjusted win rate of only 43.07% over PPO. This outcome arises because, although the model explores a wide range between $\pi_{ref}$ and $\pi_{other}$, it ultimately converges on a lower-reward region near $\pi_{other}$. These findings support our hypotheses on reward dynamics (Section 4.2) and the effects of KL divergence (Section A.4).

To explore alternatives to SALSA, we experimented with a different loss function, $\mathcal{L}_{\mathcal{MKL}}$, shown in equation 5, which regularizes the policy by averaging the KL divergences between the policy $\pi_\theta$ and two SFT models, $\pi_{\text{ref}}$ and $\pi_{\text{other}}$.

$$\mathcal{L}_{\mathcal{MKL}}(\pi_\theta) = -R(x,y) + \frac{\beta}{2}\left[\text{KL}\left(\pi_\theta, \boldsymbol{\pi}_{\text{ref}}\right) + \text{KL}\left(\pi_\theta, \pi_{\text{other}}\right)\right] \tag{5}$$

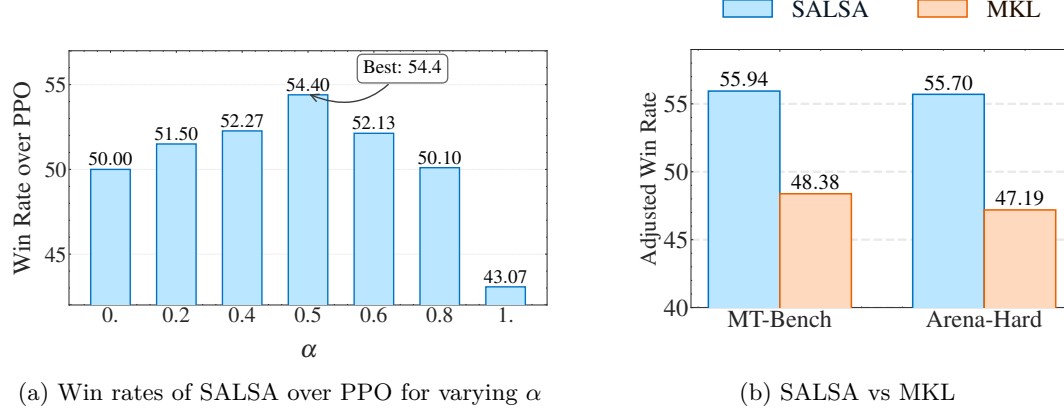

(a) Win rates of SALSA over PPO for varying $\alpha$        (b) SALSA vs MKL

Figure 4: (a) Win rates of SALSA vs. PPO (Mistral-7B) on Arena-Hard for various $\alpha$ values. (b) Win rates of SALSA and Multiple KLs over SFT (Mistral-7B) on MT-Bench and Arena-Hard.

While SALSA employs model soup (an ensemble average of two models) as a reference point, this alternative approach calculates individual divergences from each model separately. Our empirical results, as demonstrated in Figure 9d and Table 5, indicate that this method does not outperform PPO. These findings highlight the significance of the averaging methodology between SFT models: using the averaged model as a single reference point for KL divergence proves more effective than computing the average of two separate KL divergences with distinct reference points.

Finally, we conducted an ablation study to assess the impact of using soups with more than two SFT models. As illustrated in Figure 2c, the reward is higher at the midpoints between SFTs compared to the vertices. Moreover, the reward near the center of the triangle is at its peak, higher than other regions. This suggests that incorporating more SFTs into the soup is likely to enhance SALSA's performance. Figure 5 supports this hypothesis, showing that the win rate increases as the number of SFTs in the soup model grows. Although increasing the number of SFTs for constructing the soup yields better performance, we limit the number to two due to computational constraints. Investigating soups of more than three elements and finding the optimal one is left as future work.

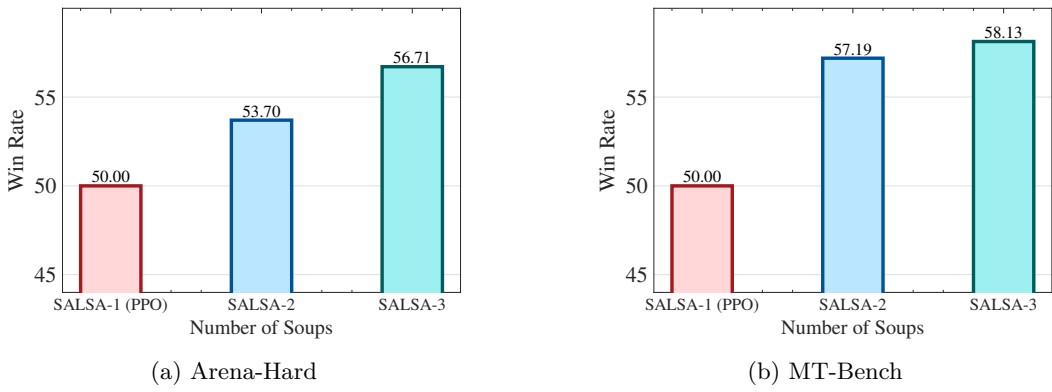

(a) Arena-Hard             (b) MT-Bench

Figure 5: Effect of the number of models in the soup on win rate. SALSA-n represents $n$ references in the soup, with SALSA-1 being equivalent to PPO. Llama2-7B is used for the above experiments.

## 5 CONCLUSION

This paper presents SALSA, a novel method for improving alignment in large language models by leveraging a model soup as a reference in the RLHF framework. By utilizing weight-space averaging of fine-tuned models as the reference, SALSA facilitates more effective exploration during policy optimization, leading to stronger performance in in-distribution and more resilience in out-of-distribution regimes. We showed that model soup resides in a higher reward region even before the PPO process, enabling SALSA to search for higher potential model. Furthermore we showed using model soup as a reference model allows for larger deviation in KL enabling search in a larger region. Experimental results across multiple benchmarks consistently show that SALSA outperforms PPO, yielding higher win rates, increased average rewards, and improved alignment with human preferences. We further extended our work to show averaging over more SFT models results even in higher win rates and robustness.

There are many avenues to extend this work: applying model soups to other forms of learning from human feedback like DPO is a very interesting future work. Systematically exploring other forms of ensembling different models as reference, and model averaging with a non-uniform or adaptive weights is another valuable line of work. Finding out remedies for KL-Hacks when using SALSA is another direction for theoretical and emperical research.

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

# A   Hyper-parameter setting

## A.1   Supervised fine-tuning setup

We use a learning rate of $2.5e - 5$, warmup ratio of 0.03, and batch size of 8 for Llama2-7B and Mistral-7B, and 16 for Gemma-2B. For each of these models, we train two supervised fine-tuned models with different random seeds. Flash attention is applied to conserve resources, and the maximum sequence length is set to 2048 tokens for all models.

## A.2   Reward model training

The reward model training uses a learning rate of $2e - 5$, with a batch size of 8. We apply a weight decay of 0.001 and use the AdamW optimizer. The learning rate schedule follows a linear decay policy.

## A.3   RLHF setup

For the RLHF stage, we use different learning rates: $2e - 6$ for Llama2-7B and Mistral-7B, and $1e - 7$ for the Gemma-2B model. The batch size is set to 8, with 2 epochs of training. We set the initial KL divergence coefficient to 0.2 for the PPO experiments and 0.01 for the SALSA experiments.

## A.4   KL divergence

The model soup resides on the line between two SFT models. As previous work by Wortsman et al. (2022) has shown, each point along the line between two fine-tuned models can improve the accuracy compared to the individual fine-tuned models. With this intuition, we hypothesize that there is a broader search space around the model soup that can still be beneficial for RLHF. Essentially, instead of searching for a model that does not diverge from a single SFT model, we can find models that do not diverge from the line between two SFT models. This allows for higher KL-divergence from the model soup. To verify this hypothesis, we tried using a small KL-divergence coefficient of $\beta = 0.01$ for PPO as well (in comparison to $\beta = 0.2$ in the optimal setting). Figure 6 illustrates that while PPO converged to a state producing gibberish output, SALSA, using the same $\beta$ value, achieved its optimal win rate.

Furthermore, high KL coefficients cannot be applied when using SALSA because the response length tends to converge to zero. Specifically, if the response length $y$ for a given prompt $x$ is zero, the KL divergence between the trained policy and the frozen policy (whether reference or soup model) also becomes zero. As a result, the policy can gravitates toward regions where the response length is zero, a phenomenon we refer to as the KL-Hack. Investigating this issue is an interesting future research direction.

---

**A gibberish response from PPO when KL coefficient is small (0.01)**

<|user|>
yo GPT can you make a text about spaghetti and sand?
<|assistant|>
Gucci Piperoni, gccce, sfkbyn Kreczel,
yucca, sand, fennel, clove, marjantrest,
yeti, sem, kava, haly, cilantro,
yeti, sem, kava, haly, clouro",
yucca, sand, fennel.

Figure 6: A gibberish response from PPO when KL coefficient is small (0.01)

## B  Prompt Used for Judgment by GPT4-Turbo

Figure 7 illustrates the prompt utilized to employ GPT4-Turbo as judge in our experiments.

---

**Prompt for Judgment**

Please act as an impartial judge and evaluate the quality of the responses provided by two AI assistants to the user question displayed below.

You should choose the assistant that follows the user's instructions and answers the user's question better. Your evaluation should consider factors such as the helpfulness, relevance, accuracy, depth, creativity, and level of detail of their responses.

Begin your evaluation by comparing the two responses and provide a short explanation. Avoid any position biases and ensure that the order in which the responses were presented does not influence your decision.

Do not allow the length of the responses to influence your evaluation. Do not favor certain names of the assistants. Be as objective as possible.

After providing your explanation, output your final verdict by strictly following this format: "[[A]]" if assistant A is better, "[[B]]" if assistant B is better, and "[[C]]" for a tie.

---

Figure 7: Instruction prompt for AI response evaluation

## C  An example of generated answers by SALSA and PPO

Figure 8 displays responses generated by SALSA and PPO for a question from the Ultra-Feedback dataset.

---

**Comparison of PPO and SALSA Methods**

**Question**

I am planning a vacation and need your help to optimize the trip

| **PPO** | **SALSA** |
|---|---|
| I do not have access to information about your specific vacation goals and interests. However, here are some general tips that can help you optimize your vacation: 

 1. Choose a travel package: there are many travel packages that include a range of services, including flights, hotel stays, and activities. Choose one that includes everything you need to have a comfortable and enjoyable vacation. 

 2. Book your flights and hotel stays ahead of time: booking your flights and hotel stays ahead of time can save you money and time. It also gives you a better chance to get the best deals and availability. 3. Book your activities in advance: booking your activities in advance can save you money and time. Many attractions and activities have discounts or promotions for booking ahead of time. 

 4. Use public transportation: taking public transportation is a great way to save money on transportation costs. It also allows you to experience the local culture and get off the beaten path. 

 5. Eat like a local: eating like a local can save you money on eating out and also give you a taste of the local culture. | Sure, I'd be happy to help you optimize your vacation! Here are some tips to get you started: 

 1. Plan ahead: Take some time to plan your vacation in advance, including your travel dates, destinations, and activities. This will help you avoid last-minute expenses and ensure that you have enough time to do everything you want to do. 

 2. Research your destination: Do your research on your destination, including what you want to see, do, and eat. This will help you make the most of your time and money. 

 3. Use travel apps: There are many travel apps that can help you stay organized and on track. For example, TripAdvisor can help you find the best restaurants and attractions, while Google Maps can help you plan your transportation. 

 4. Make use of travel rewards: Take advantage of your travel rewards, including points and miles. This will help you save money on your trip and make it more enjoyable. 

 5. Be flexible: Be flexible with your travel plans, and don't be afraid to try new things. This will help you avoid getting stuck in a routine and make your trip more exciting. |

---

Figure 8: Qualitative comparison of PPO and SALSA.

Table 2: Comparison of SALSA, PPO, and SFT across Various Datasets for Llama2-7B Model

| Dataset | Comparison | Win | Loss | Tie | Win Rate | Adj. Win Rate |
|---|---|---|---|---|---|---|
| **Llama2-7B Model** | | | | | | |
| **MT-Bench** | SALSA | 29 | 21 | 110 | **18.12** | **52.50** |
| | PPO | 21 | 29 | 110 | 13.12 | 47.50 |
| | SALSA | 36 | 28 | 96 | **22.50** | **52.50** |
| | SFT | 28 | 36 | 96 | 17.50 | 47.50 |
| | PPO | 23 | 21 | 116 | 14.37 | 50.63 |
| | SFT | 21 | 23 | 116 | 13.12 | 49.37 |
| **Arena-Hard** | SALSA | 102 | 62 | 335 | **20.44** | **54.01** |
| | PPO | 62 | 102 | 335 | 12.42 | 45.99 |
| | SALSA | 91 | 44 | 365 | **18.20** | **54.70** |
| | SFT | 44 | 91 | 365 | 8.80 | 45.30 |
| | PPO | 58 | 52 | 387 | 11.67 | 50.60 |
| | SFT | 52 | 58 | 387 | 10.46 | 49.40 |
| **UltraFeedback** | SALSA | 477 | 447 | 1075 | **23.86** | **50.75** |
| | PPO | 447 | 477 | 1075 | 22.36 | 49.25 |
| | SALSA | 504 | 415 | 1080 | **25.21** | **52.23** |
| | SFT | 415 | 504 | 1080 | 20.76 | 47.77 |
| | PPO | 443 | 417 | 1136 | 22.19 | 50.65 |
| | SFT | 417 | 443 | 1136 | 20.89 | 49.35 |

## D  WIN RATES

This section provides a comprehensive breakdown of win rates across all our experiments. We have included the details of win rates for all the experiments for llama2-7b in table 2, for gemma-2b in table 4 and for Mistral-7b in table 3. For each dataset we have compared the win rates of SALSA over PPO, SALSA over SFT and PPO over SFT. We have included win rates and adjusted win rates. SALSA consistently outperforms PPO and SFT.

## E  MULTIPLE KL

Table 5 gives additional info of win MKL win rate over PPO. We compared MKL win rate over PPO over MT-Bench and Arena-Hard and in both cases MKL doesn't outperform PPO. This means simply trying to use multiple models is not gonna be effective and doing weight averaging in SALSA is crucial for effectiveness.

## F  KL TRAINING CURVES

In this section we have included the KL and rewards for ultra feedback dataset. In figure 9 the KL curves and rewards of PPO and SALSA look similar, KL starts to get some distance from the original point in SALSA similar to PPO. Reward on samples seen during training for both methods increases similarly. The final model of SALSA has higher reward over the whole dataset as it can bee seen in Fiugre 3.

Table 3: Comparison of SALSA, PPO, and SFT across Various Datasets for Mistral-7B model

| Dataset | Comparison | Win | Loss | Tie | Win Rate | Adj. Win Rate |
|---|---|---|---|---|---|---|
| **Mistral-7B Model** | | | | | | |
| | SALSA | 30 | 24 | 105 | 18.87 | 51.89 |
| | PPO | 24 | 30 | 105 | 15.09 | 48.11 |
| **MT-Bench** | SALSA | 42 | 23 | 95 | 26.25 | 55.94 |
| | SFT | 23 | 42 | 95 | 14.37 | 44.06 |
| | PPO | 28 | 23 | 109 | 17.50 | 51.56 |
| | SFT | 23 | 28 | 109 | 14.37 | 48.44 |
| | SALSA | 109 | 65 | 326 | 21.80 | 54.40 |
| | PPO | 65 | 109 | 326 | 13.00 | 45.60 |
| **Arena-Hard** | SALSA | 126 | 69 | 305 | 25.20 | 55.70 |
| | SFT | 69 | 126 | 305 | 13.80 | 44.30 |
| | PPO | 71 | 67 | 362 | 14.20 | 50.40 |
| | SFT | 67 | 71 | 362 | 13.40 | 49.60 |
| | SALSA | 497 | 401 | 1102 | 24.85 | 52.40 |
| | PPO | 401 | 497 | 1102 | 20.05 | 47.60 |
| **UltraFeedback** | SALSA | 465 | 388 | 1146 | 23.26 | 51.93 |
| | SFT | 388 | 465 | 1146 | 19.41 | 48.07 |
| | PPO | 455 | 450 | 1095 | 22.75 | 50.65 |
| | SFT | 450 | 455 | 1095 | 20.89 | 49.35 |

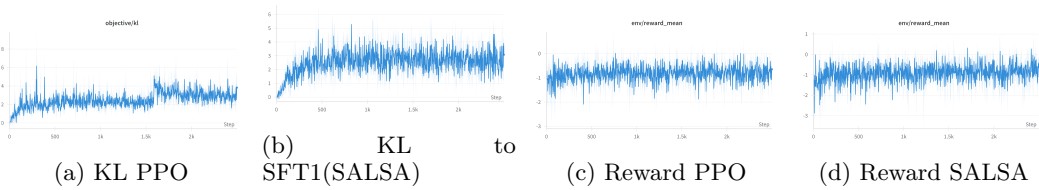

(a) KL PPO    (b) KL to SFT1(SALSA)    (c) Reward PPO    (d) Reward SALSA

Figure 9

Table 4: Comparison of SALSA, PPO, and SFT across Various Datasets for Gemma-2B model

| Dataset | Comparison | Win | Loss | Tie | Win Rate | Adj. Win Rate |
|---------|-----------|-----|------|-----|----------|---------------|
| | **Gemma-2B Model** | | | | | |
| | SALSA | 37 | 14 | 109 | **23.13** | **57.19** |
| | PPO | 14 | 37 | 109 | 8.75 | 42.81 |
| **MT-Bench** | SALSA | 40 | 18 | 102 | **25.0** | **56.88** |
| | SFT | 18 | 40 | 102 | 11.25 | 43.12 |
| | PPO | 14 | 8 | 138 | 8.75 | 51.88 |
| | SFT | 8 | 14 | 138 | 5.0 | 48.12 |
| | SALSA | 154 | 117 | 229 | **30.8** | **53.7** |
| | PPO | 117 | 154 | 229 | 23.4 | 46.3 |
| **Arena-Hard** | SALSA | 154 | 116 | 230 | **30.8** | **53.8** |
| | SFT | 116 | 154 | 230 | 23.2 | 46.2 |
| | PPO | 24 | 19 | 451 | 4.86 | 50.51 |
| | SFT | 19 | 24 | 451 | 3.84 | 49.49 |
| | SALSA | 174 | 147 | 1660 | **8.78** | **50.68** |
| | PPO | 147 | 174 | 1660 | 7.42 | 49.32 |
| **UltraFeedback** | SALSA | 300 | 129 | 1548 | **15.17** | **54.32** |
| | SFT | 129 | 300 | 1548 | 6.53 | 45.68 |
| | PPO | 276 | 129 | 1580 | 13.90 | 53.70 |
| | SFT | 129 | 276 | 1580 | 6.50 | 46.30 |

Table 5: Comparison of MKL and PPO across MT-Bench and Arena-Hard for Mistral-7B.

| Dataset | Comparison | Win | Loss | Tie | Win Rate | Adj. Win Rate |
|---------|-----------|-----|------|-----|----------|---------------|
| **MT-Bench** | MKL | 69 | 85 | 340 | 13.97 | 48.38 |
| | PPO | 85 | 69 | 340 | 17.21 | **51.62** |
| **Arena-Hard** | MKL | 20 | 29 | 111 | 12.50 | 47.19 |
| | PPO | 29 | 20 | 111 | 18.13 | **52.81** |

