# OpenReview forum: "SALSA: Soup-based Alignment Learning for Stronger Adaptation in RLHF"
_ICLR.cc/2025/Conference — Submitted to ICLR 2025_

### Official Review · Reviewer_yT8z · 2024-11-04

**Soundness:** 3
**Presentation:** 3
**Contribution:** 2
**Rating:** 5
**Confidence:** 3

**Summary:**

This work investigates improving the reference model in RLHF through model merging. The authors propose to use the average parameter of several SFT models that are initialized from the same base model but trained with different seeds as the reference model in the PPO algorithm. Their results demonstrate that by using a "soup" model as the reference model, PPO obtains a significant performance improvement in three dialogue benchmarks.

**Strengths:**

1. The proposed technique is clear and effective according to the experiment results.
2. The proposed technique is evaluated with different base models.
3. The authors provide sufficient analysis of the impact of model merging in the final reward, which well motivates the proposed technique.

**Weaknesses:**

1. As important algorithms in RLHF, reward-free methods such as DPO are not discussed in this paper.
2. The proposed technique is not fully explored in the context of RLHF. For example, the soup model can also serve to be the reference model in reward-free RLHF methods like DPO.

**Questions:**

1. How does the SALSA perform when compared to DPO?
2. Can using the soup model as a reference model also bring benefit to DPO?
3. In the implementation of SALSA, how is the model initialized at the beginning of the RLHF process? If the soup model is used for initialization, a further analysis about the impact of using soup model as the RL initialization model would be expected.
4. Line-321: "high KL coefficients cannot be applied when using SALSA because the response length tends to converge to zero. Specifically, if the response length y for a given prompt x is zero, the KL divergence between the trained policy and the frozen policy also becomes zero." This statement is incorrect since when the response length y is zero, the EOS token would contribute a non-zero value for KL divergence and thus the KL divergence between the trained policy and the frozen policy would not be zero.

---

> ### Author Response · Authors · 2024-11-22
> **Reviewer response**
>
> Thanks for showing interest and in our work and finding it clear and effective. Here are our responses to your concerns:
>
> >As important algorithms in RLHF, reward-free methods such as DPO are not discussed in this paper.
> > The proposed technique is not fully explored in the context of RLHF. For example, the soup model can also serve to be the reference model in reward-free RLHF methods like DPO.
>
> SALSA can be combined with DPO, so it is not a direct comparison. We have compared the case when chose model soup as reference model in PPO. Doing the same experiment with DPO is expected to improve the results, but it is orthogonal to our work and considered a future work.
>
> > How does the SALSA perform when compared to DPO?
> SALSA can be combined with DPO, so it is not a direct comparison. We have compared the case of using model soup as reference model in PPO and have shown improvements there. Doing the same experiment with DPO is expected to improve the results, but it is orthogonal to our work and considered a future work.
>
> > Can using the soup model as a reference model also bring benefit to DPO?
> Yes we believe that using it in DPO should also improve. The reason is the model soup resides in a higher reward space, as it can be seen in Figure 2.c. This is happening before any RLHF process is applied.
>
> > In the implementation of SALSA, how is the model initialized at the beginning of the RLHF process? If the soup model is used for initialization, a further analysis about the impact of using soup model as the RL initialization model would be expected.
>
> The model is initialized as SFT1 model, the model soup is only used for computing KL divergence in the following equation:
> $\mathcal{L_{SALSA}}(\pi_{\theta}) = -R(x, y) + \beta \text{KL}\left(\pi_{\theta}(y \mid x) \|\| {\boldsymbol{\pi_\text{{soup}}}}(y \mid x)\right)$
>
>
>
>
> [1] Unpacking DPO and PPO  https://arxiv.org/html/2406.09279v1
>
> [2] Is DPO superior to PPO in alignment?  https://arxiv.org/html/2404.10719v1
>
> [3] Back to basics https://arxiv.org/abs/2402.14740

---

> > ### Comment · Reviewer_yT8z · 2024-11-25
> >
> > Thanks for the response. The authors don't solve my concern, so i would like to keep my score as it is.

---

> ### Author Response · Authors · 2024-12-03
> **Response to reviewer**
>
> It seems we had slipped to put the answer for your last question here. There were some editing issues for your other questions too, (some of our answers were merged to the questions and one was missing).  We are extremely sorry about this inconvenience, it was a last minute edit before going to thanksgiving break so we didn't catch it. (You can even see there were some references without being used in the text which was a editing issue) Copying all the answers here with correct editing format:
>
> Thanks for showing interest in our work and finding it clear and effective. Here are our responses to your concerns:
>
> >As important algorithms in RLHF, reward-free methods such as DPO are not discussed in this paper.
>
> We considered PPO because [1,2] had shown PPO is superior to DPO. We also found out in our internal experiments PPO is more reliable. Applying the same techniques on DPO, or using other algorithms such as Reinforce as suggested by [3] are considered a future work.
>
> > The proposed technique is not fully explored in the context of RLHF. For example, the soup model can also serve to be the reference model in reward-free RLHF methods like DPO.
>
> > How does the SALSA perform when compared to DPO?
>
> SALSA can be combined with DPO, so it is not a direct comparison. We have compared the case of using model soup as reference model in PPO and have shown improvements there. Doing the same experiment with DPO is expected to improve the results, but it is orthogonal to our work and considered a future work.
>
> > Can using the soup model as a reference model also bring benefit to DPO?
>
> Yes we believe that using it in DPO should also improve. The reason is the model soup resides in a higher reward space, as it can be seen in Figure 2.c. This is happening before any RLHF process is applied.
>
> > In the implementation of SALSA, how is the model initialized at the beginning of the RLHF process? If the soup model is used for initialization, a further analysis about the impact of using soup model as the RL initialization model would be expected.
>
> The model is initialized as SFT1 model, the model soup is only used for computing KL divergence in the following equation:
> $\mathcal{L_{SALSA}}(\pi_{\theta}) = -R(x, y) + \beta \text{KL}\left(\pi_{\theta}(y \mid x) \|\| {\boldsymbol{\pi_\text{{soup}}}}(y \mid x)\right)$
>
> > Line-321: "high KL coefficients cannot be applied when using SALSA because the response length tends to converge to zero. Specifically, if the response length y for a given prompt x is zero, the KL divergence between the trained policy and the frozen policy also becomes zero." This statement is incorrect since when the response length y is zero, the EOS token would contribute a non-zero value for KL divergence and thus the KL divergence between the trained policy and the frozen policy would not be zero.
>
> You are actually right that in never becomes 0 exactly. We have mentioned in the paper that it tends to converge to 0, it doesn't become exactly 0. When we have lower number of tokens, the loss will become much less (in case of EOS token it is only 1 token). We will rewrite that section to clarify this ambiguity. SIMPO [4] actually has done length normalization to make rewards more related to quality of response rather than the length. Basically having more tokens will add to KL divergence as in the  $log(P(y|x)/Q(y|x)) = \sum_{i} log(P(y_i|x)/Q(y_i|x))  $ the sum will increase. If all tokens were iid 1 token to 100 tokens have roughly 1/100 th loss which is why we say gets close to 0. They are not exactly iid but it is getting smaller w.r.t to length.
>
>
> [1] Unpacking DPO and PPO  https://arxiv.org/html/2406.09279v1
>
> [2] Is DPO superior to PPO in alignment?  https://arxiv.org/html/2404.10719v1
>
> [3] Back to basics https://arxiv.org/abs/2402.14740
>
> [4] SIMPO https://arxiv.org/abs/2405.14734
>
> We hope these answers will clarify your questions and help increase the score. Again extremely sorry about the editing issue happened.

---

### Official Review · Reviewer_NSBF · 2024-11-04

**Soundness:** 3
**Presentation:** 3
**Contribution:** 2
**Rating:** 3
**Confidence:** 4

**Summary:**

The paper introduces SALSA (Soup-based Alignment Learning for Stronger Adaptation), a novel method in the field of large language models, aimed at enhancing model alignment through Reinforcement Learning from Human Feedback (RLHF). Unlike traditional methods that rely on a fixed initial policy as a reference, SALSA employs a model soup created by weight-space averaging of two independently fine-tuned models. This approach allows for greater flexibility and exploration during policy optimization, resulting in improved model performance and robustness.

**Strengths:**

The SALSA method introduces the novel concept of model soup to address alignment issues in large language models. By employing weight-space averaging of two independently fine-tuned models, this innovation overcomes the limitations of KL divergence constraints in traditional RLHF methods, opening new research directions.
The paper is well-structured and logically coherent, offering detailed and easily understandable descriptions of the SALSA method. By contrasting traditional methods with SALSA, the authors effectively convey the core ideas and advantages of the method, enabling readers to clearly grasp its contributions.

**Weaknesses:**

1 Limited Dataset Generality: The experiments primarily use the UltraChat-200k and UltraFeedback datasets, which may not be comprehensive enough to reflect the model's performance across different domains or applications. As a result, the findings might not be generalizable to broader real-world scenarios.

2 Lack of Training Curve Presentation: The article does not provide training curves, making it difficult for readers to assess the model's convergence and stability during training. Training curves can help understand the learning dynamics and potential issues in the optimization process.

3 Lack of Comparison with Other Related Works: Apart from PPO, the article does not compare the SALSA method with other existing alignment or optimization methods. This limits a comprehensive understanding of SALSA's relative advantages and its competitiveness in a broader context.

Reviewers will consider raising the score if the author can effectively address their concerns.

**Questions:**

Here are some questions and suggestions for the authors:

Dataset Generality:Could you provide more details on the choice of datasets? How do you ensure that the results are generalizable across different domains?

Have you considered testing the model on additional datasets to validate its performance more broadly?

Training Curves:Can you include training curves in your supplementary materials? These would help in understanding the convergence behavior and stability of your model.What insights can you provide about the model's learning dynamics during training?

Comparison with Other Methods:Why did you choose to compare only with PPO? Are there specific reasons for not including other related methods in your analysis?Would you consider conducting experiments with other alignment or optimization methods to provide a more comprehensive evaluation?

KL-Hack Issue:Can you elaborate on the KL-Hack phenomenon and potential strategies to mitigate it?Have you explored any preliminary solutions or adjustments to address this issue?

Future Work and Extensions: What are your plans for extending this research? Are there specific areas you believe are promising for further exploration?How do you envision incorporating more than three SFT models into the model soup, considering computational constraints?

These questions aim to clarify certain aspects of your work and suggest areas for potential improvement or further research.

---

> ### Author Response · Authors · 2024-11-22
> **Reviewer response**
>
> > Dataset Generality:Could you provide more details on the choice of datasets? How do you ensure that the results are generalizable across different domains?
>
> We followed SIMPO[1]’s dataset choice for doing RLHF process as a well respected paper within the community. We chose UltraFeedback for training reward model, but we have tested it on MT-Bench and Arena-hard too.
>
> >  Have you considered testing the model on additional datasets to validate its performance more broadly?
>
> The experiments primarily use the UltraChat-200k and UltraFeedback datasets, which may not be comprehensive enough
> We have tested the model on MT-Bench and Arena-hard too. These datasets are considered out of distribution for Ultrafeedback, therefore we have evaluated both in distribution and out of distribution.
>
> >Training Curves:Can you include training curves in your supplementary materials? These would help in understanding the convergence behavior and stability of your model.What insights can you provide about the model's learning dynamics during training?
>
> We will include these curves in the camera ready version of the paper. We made sure KL divergence looks like standard RLHF results, no negative KL happens, results of RLHF are sensible and other standard tests. The results of PPO on KL and SALSA on KL were looking similar.
>
> > Comparison with Other Methods:Why did you choose to compare only with PPO? Are there specific reasons for not including other related methods in your analysis?Would you consider conducting experiments with other alignment or optimization methods to provide a more comprehensive evaluation?
>
> We considered PPO because [2,3] had shown PPO is superior to DPO. We also found out in our internal experiments PPO is more reliable. Applying the same techniques on DPfO, or using other algorithms such as Reinforce as suggested by [4] are considered a future work.
>
> > KL-Hack Issue:Can you elaborate on the KL-Hack phenomenon and potential strategies to mitigate it?Have you explored any preliminary solutions or adjustments to address this issue?
>
> We are using model soup as reference model in this equation :
> $\mathcal{L_{SALSA}}(\pi_{\theta}) = -R(x, y) + \beta \text{KL}\left(\pi_{\theta}(y \mid x) \|\| {\boldsymbol{\pi_\text{{soup}}}}(y \mid x)\right)$
>
> In this case when KL coefficient Beta is high, the model tends to hack KL divergence and provide zero length output. The reason this happens is the $\pi_\theta$ model that starts to be optimized is far from model soup, and when we push it to become too close to model soup it can’t find a solution and hacks the KL instead. Therefore small betas work best for SALSA. We have mentioned this as a disadvantage of SALSA and can be considered as a future work to improve it.
>
> > Future Work and Extensions: What are your plans for extending this research? Are there specific areas you believe are promising for further exploration?How do you envision incorporating more than three SFT models into the model soup, considering computational constraints?
>
> There are a few directions that can be considered:
> * Applying same technique to other RLHF methods, such as DPO, Reinforce, etc.
> * Resolving the KL hack issue to make SALSA work with higher betas too.
> * how the algorithms like TRPO and PPO can be improved by using multiple points rather one
>
> [1] SimPO https://arxiv.org/abs/2405.14734
>
> [2] Unpacking DPO and PPO  https://arxiv.org/html/2406.09279v1
>
> [3] Is DPO superior to PPO in alignment?  https://arxiv.org/html/2404.10719v1
>
> [4] Back to basics https://arxiv.org/abs/2402.14740

---

> > ### Author Response · Authors · 2024-11-22
> > **training curve**
> >
> > We added some training curves to appendix per your request.

---

> > ### Comment · Reviewer_NSBF · 2024-11-30
> >
> > I appreciate the authors' effort in providing detailed KL and reward curves during training (Fig.9), which offer helpful additional insights. However, the training curve of SALSA does not show substantial differences compared to the baselines. I have decided to maintain my original score.

---

> ### Author Response · Authors · 2024-12-03
> **New Response to reviewer**
>
> Hi, It seems that there was some formatting issue with our previous response, and some of the responses were merged into questions. Extremely sorry about editing issues. We edited the response and our answer should be distinguishable from your questions now, Could you please read it again? We kindly ask to improve your score as we had answered all of your questions.
>
> Regarding new questions:
> > However, the training curve of SALSA does not show substantial differences compared to the baselines.
>
> First of all reward in training curve for SALSA in figure 9d is visibly better than PPO in 9c.  (For  SALSA it is close to -0.25 in average last .5k steps but for PPO close to -0.75 in average last .5k steps).
>
> Secondly, using lower KL divergence coefficient for PPO will result in explosion. For the KL divergence for both PPO and SALSA we tried to use  the lowest possible beta to get the best reward possible. In figure 9a (PPO), it is visible it already jumps after 1.5k steps. If we made the beta smaller than this KL would have jumped significantly, In figure 9b (SALSA) the KL divergence doesn't jump even though smaller beta has been used. It was possible to go for lower betas for SALSA, because it resides in a region surrounded by high reward models. This phenomenon can be seen Figure 2.c; there is a big circle around average models that are considered to be good models, but the same idea can't apply to PPO which are at the corners of the triangle.
>
> Thirdly, since the beta is used to get the best outcome for each of them it was a fair comparison.

---

### Official Review · Reviewer_K7rw · 2024-11-05

**Soundness:** 3
**Presentation:** 2
**Contribution:** 3
**Rating:** 6
**Confidence:** 4

**Summary:**

The authors propose an alternative to RLHF which instantiates two models and then takes the average of them at each training step, replacing the KL divergence term in the classical RLHF setup.

**Strengths:**

- interesting and nice train of work!
- I appreciate the novel look at RLHF.

**Weaknesses:**

- PPO at this point has fallen out of favor as a canonical RLHF architecture (see the Back to Basics paper by Cohere AI). Furthermore, PPO operates using a trust region policy, so it bounds the steps that a learning update could anyway. In order for me to recommend your paper for publication, I'd want to see you repeat this experiment with REINFORCE, Actor-Critic, and perhaps another RL algorithm that makes up the backbone of your RLHF pipeline.

**Questions:**

- Is your model soup similar to a model ensemble? I’m curious as to why there is the separate terminology.

---

> ### Author Response · Authors · 2024-11-22
> **Response to reviewer**
>
> Thanks for for finding our work interesting and novel. Here are our response to your concerns, please let us know if any further discussion is required.
>
> > PPO at this point has fallen out of favor as a canonical RLHF architecture (see the Back to Basics paper by Cohere AI). Furthermore, PPO operates using a trust region policy, so it bounds the steps that a learning update could anyway. In order for me to recommend your paper for publication, I'd want to see you repeat this experiment with REINFORCE, Actor-Critic, and perhaps another RL algorithm that makes up the backbone of your RLHF pipeline.
>
> We appreciate reviewer’s mentioning how RLHF can use other algorithms for improvement. Our paper is focusing on which regions have higher reward models rather than the RL algorithm itself. The definition of reward can change in each of those methods to SALSA’s definition of reward. As it can be seen in Figure 2.c rewards around model soup is higher than original model even before RLHF process. Therefore we expect the same reward gains to happen in other optimization methods too. On another note, [1,2] have shown PPO is superior to DPO and have considered PPO is one of the most popular preference based optimization algorithms. Also [3,4] which are more recent than the "Back to basics" paper still consider PPO a popular option, so within the community it is still a popular approach. Unfortunately, rerunning with the new algorithms takes much more time than the discussion period, but hopefully our reasoning here would justify the use case.
>
> > Is your model soup similar to a model ensemble? I’m curious as to why there is the separate terminology.
>
> This is a very great question, SALSA averages the weights of the models and use them as reference point. Model ensembles on the other hand average the probabilities of two models for obtaining output. The most close experiment of comparing SALSA with model ensemble is Figure 5.b. In this figure a method named MKL which gets the average of KL w.r.t two SFT points is compared to SALSA. SALSA outpeforms MKL with significant margin. The improvement of SALSA comes from the fact it resides in a higher reward space as shown in Figure 2.c. But in a model ensemble both points resides in low reward region, therefore we don't expect it to improve the results.
>
>
> [1] Unpacking DPO and PPO  https://arxiv.org/html/2406.09279v1
>
> [2] Is DPO superior to PPO in alignment?  https://arxiv.org/html/2404.10719v1
>
> [3] Understanding the performance gap between online and offline alignment algorithms.  https://arxiv.org/abs/2405.08448
>
> [4] SimPO: Simple Preference Optimization with a Reference-Free Reward https://arxiv.org/abs/2405.14734

---

> > ### Comment · Reviewer_K7rw · 2024-12-03
> >
> > Hey, thanks for the response to my question.
> >
> > I do think that using REINFORCE instead of a PPO may provide a more competitive baseline. I.e. maybe improvements are marginal when using a more "powerful" RLHF algorithm.  Furthermore, and this is just my opinion, but many practitioners I've seen consider DPO to be weak as well at this point, so just because PPO is better than it doesn't mean that PPO is a competitive baseline at this point.
> >
> > I've increased my scores. Best of luck guys.

---

> > > ### Author Response · Authors · 2024-12-04
> > > **Response**
> > >
> > > We would like to thank the reviewer for increasing their score.

---

### Official Review · Reviewer_T2cX · 2024-11-06

**Soundness:** 2
**Presentation:** 3
**Contribution:** 2
**Rating:** 5
**Confidence:** 4

**Summary:**

The paper proposes a simple modification to the KL-constrained objective used in RL finetuning. Instead of regularizing to the same fixed policy $\pi_{ref}$, the authors propose to regularize to a souped policy $\pi_{soups}$ which is the model soups of multiple $\pi_{ref}$ (each SFT-ed on the same dataset with different seed or hyper-parameters). The authors claim this is a more robust ref model, and demonstrate results on instruction following benchmarks (MT-Bench, Arena-Hard, UltraFeedback) where the proposed approach improves over the classic $\pi_{ref}$ regularization.

**Strengths:**

1. The paper illustrates a simple phenomenon, that applying model soups to the anchor policy for KL regularization can lead to improved win-rates in many benchmarks.
2. The paper includes some ablations, including on how the policies are souped (finding that uniform souping is best). Also, the authors show that multiple KL terms to each individual policy does not lead to any benefits, whereas a single KL term to a souped policy does.

**Weaknesses:**

1. I think some claims made in the paper are not rigorously justified. For example, Figure 2 shows that reward increases with model souping, but it is also important to plot the KL wrt a fixed anchor policy (whether pi_ref or pi_soups or both). In other words, it may be possible to get a higher reward by simply having higher KL and this is not ruled out in the experiments.
2. Related to (1), simply showing higher win-rates in Table 1, without plotting the KL as well, is also not very convincing, since these benchmarks are easy to game. It would be much more convincing if SALSA leads to higher rewards/win-rate while maintaining the same or lower KL compared to pi_ref.
3. The paper does not discuss any important related works that apply model soups to RLHF, for example:
[1] "WARP: On the Benefits of Weight Averaged Rewarded Policies", https://arxiv.org/abs/2406.16768
[2] "WARM: On the Benefits of Weight Averaged Reward Models", https://arxiv.org/abs/2401.12187
[3] "Conditional Language Policy: A general framework for steerable multi-objective finetuning", https://arxiv.org/abs/2407.15762

**Questions:**

1. Could the authors plot results showing also KL wrt fixed anchor policy? (as mentioned in Weaknesses)
2. In the abstract and several places in the paper, the author claimed that model soups lead to "improving model robustness, out-of-distribution generalization". How are model robustness and OOD generalization entailed by the experiments?

---

> ### Author Response · Authors · 2024-11-22
> **Response to the reviewer**
>
> Thanks for finding our work interesting and asking questions in details. Here is our response and we hope we can discuss further to resolve any concerns:
>
> > I think some claims made in the paper are not rigorously justified. For example, Figure 2 shows that reward increases with model souping, but it is also important to plot the KL wrt a fixed anchor policy (whether pi_ref or pi_soups or both). In other words, it may be possible to get a higher reward by simply having higher KL and this is not ruled out in the experiments.
>
> Figure 2 is showing the reward before any RLHF process as illustrated in section 4.2 (reward analysis). Basically as we get closer to average of the models, the reward gets higher.  Therefore it is completely unrelated to KL divergence coefficients.
>
> This is actually one of our main findings, not only model soups have better accuracy on the function that SFT models have been trained on, they have higher values on reward function that was independently trained.
>
> > Related to (1), simply showing higher win-rates in Table 1, without plotting the KL as well, is also not very convincing, since these benchmarks are easy to game. It would be much more convincing if SALSA leads to higher rewards/win-rate while maintaining the same or lower KL compared to pi_ref.
> For the win rate experiments and reward comparison of PPO vs SALSA we chose the best KL factor that gives highest win rate for each of PPO and SALSA, therefore the better win rates are not coming from higher KL.  (Also mentioned in ablation studies) Therefore this is not the case.
>
> >The paper does not discuss any important related works that apply model soups to RLHF, for example: [1] "WARP: On the Benefits of Weight Averaged Rewarded Policies", https://arxiv.org/abs/2406.16768 [2] "WARM: On the Benefits of Weight Averaged Reward Models", https://arxiv.org/abs/2401.12187 [3] "Conditional Language Policy: A general framework for steerable multi-objective finetuning", https://arxiv.org/abs/2407.15762
>
> Thanks for mentioning these related papers. We will definitely include them in the next version of the paper. In WARP the SFT model is weight averaged in RLHF models which is $\pi_\theta$ in our equations. But we are changing the reference point within the KL divergence ($\pi_\text{soup}$ in the loss). Therefore, it is different. WARM on the other hand is averaging reward model which is R(x,y) in our loss function, which we don't change. Hence again it is different for this case.
>
> Loss function (equation 4) from the paper for better understanding within the chat:
>
> $\mathcal{L_{SALSA}}(\pi_{\theta}) = -R(x, y) + \beta \text{KL}\left(\pi_{\theta}(y \mid x) \|\| {\boldsymbol{\pi_\text{{soup}}}}(y \mid x)\right)$
>
> > In the abstract and several places in the paper, the author claimed that model soups lead to "improving model robustness, out-of-distribution generalization". How are model robustness and OOD generalization entailed by the experiments?
>
> The UltraFeedback has been used in the reward model and preference optimization. The MT-Bench and Arena-Hard datasets are considered out of distribution for this dataset.
>
> PPO and SFT often perform similarly on MT-Bench and Arena-Hard, likely because UltraFeedback and UltraChat, which are utilized for SFT, Reward Modeling, and RLHF, are considered out-of-distribution for these benchmarks. As a result, a basic version of PPO does not significantly outperform SFT. However, SALSA's robustness to out-of-distribution data (derived from weight averaging and model soup techniques) delivers improvements of up to 57\% and 54\% on these datasets. (Mentioned in main results section)

---

> > ### Author Response · Authors · 2024-11-22
> > **KL Plots**
> >
> > We have added KL plots to appendix per your request.

---

> > ### Comment · Reviewer_T2cX · 2024-11-26
> >
> > Thanks authors for the response. I have two follow-up questions.
> >
> > > Figure 2 is showing the reward before any RLHF process as illustrated in section 4.2 (reward analysis). Basically as we get closer to average of the models, the reward gets higher. Therefore it is completely unrelated to KL divergence coefficients.
> >
> > The KL divergence matters because larger KL budget makes it much easier to optimize the reward. In the new KL plots (Figure 9), it seems that PPO gets lower KL which may explain why it obtains smaller reward. Thus to make the comparison fair, you should report both the KL value and reward values in the main text results.
> >
> > > The UltraFeedback has been used in the reward model and preference optimization. The MT-Bench and Arena-Hard datasets are considered out of distribution for this dataset.
> >
> > How is MT-Bench and Arena-Hard significantly different from UltraFeedback? From my quick scan, they all seem to be instruction following datasets so it's not clear how "out of distribution" these evals are. Can you provide some more details on this?

---

> > > ### Author Response · Authors · 2024-11-30
> > > **Response to reviewer**
> > >
> > > Thanks for reading our revision.
> > > - From the plots the final KL seems to be around 3 to 4 in both cases. The figure quality of this one was bad so it might look like so. We will improve the quality of figures and add the additional details to main text in camera ready if accepted.
> > > -  MT-bench vs ultra feedback: MT-Bench serves as a benchmark to evaluate the conversational abilities of language models, while UltraFeedback offers a comprehensive dataset aimed at improving model alignment through detailed feedback.
> > > - Arena-hard vs UltraFeedback: Arena-Hard functions as a benchmark to evaluate LLM performance on complex, open-ended tasks, while UltraFeedback serves as a dataset to train and enhance reward models, so there is less focus on complexity in UltraFeeback.
> > > In general all RLHF datasets are similar in terms of following instructions, but they are different in content. (This is the case for other kind of datasets too, for example, in computer vision CIFAR and ImageNet are both for classifying images, but the complexity and types of question can vary the datasets.)

---

> > > > ### Author Response · Authors · 2024-12-03
> > > >
> > > > Thank you for your valuable feedback and engagement! If you feel that we have sufficiently addressed your concerns, we would greatly appreciate it if you could consider reflecting this in your score.
> > > >
> > > > Should you have any additional concerns or suggestions, we would be more than happy to address them.

---

> > > > ### Author Response · Authors · 2024-12-04
> > > > **KL divergence**
> > > >
> > > > Another reviewer had a similar question to yours, copying the responses here in case it will resolve your concerns.
> > > >
> > > > > However, the training curve of SALSA does not show substantial differences compared to the baselines.
> > > >
> > > > First of all reward in training curve for SALSA in figure 9d is visibly better than PPO in 9c. (For SALSA it is close to -0.25 in average last .5k steps but for PPO close to -0.75 in average last .5k steps).
> > > >
> > > > Secondly, using lower KL divergence coefficient for PPO will result in explosion. For the KL divergence for both PPO and SALSA we tried to use the lowest possible beta to get the best reward possible. In figure 9a (PPO), it is visible it already jumps after 1.5k steps. If we made the beta smaller than this KL would have jumped significantly, In figure 9b (SALSA) the KL divergence doesn't jump even though smaller beta has been used. It was possible to go for lower betas for SALSA, because it resides in a region surrounded by high reward models. This phenomenon can be seen Figure 2.c; there is a big circle around average models that are considered to be good models, but the same idea can't apply to PPO which are at the corners of the triangle.
> > > >
> > > > Thirdly, since the beta is used to get the best outcome for each of them it was a fair comparison.
> > > >
> > > > Again thanks for your effort and hope this can resolve your concerns.

---

### Official Review · Reviewer_pSBg · 2024-11-07

**Soundness:** 3
**Presentation:** 3
**Contribution:** 2
**Rating:** 5
**Confidence:** 4

**Summary:**

This paper considers using the technology of model soup by weighted averaging two reference models which is fine-tuned on the same pre-trained model to improve the reference model in RLHF. It demonstrates that using model soup to improve the reference model will lead to better aligned model by PPO algorithm.

**Strengths:**

1. The experiment is well done. It demonstrates the advantage of model soup for reference model in RLHF from different aspects. It explores the different weighting ratio when averaging two or three models. When the ratio is averaged, the aligned model by PPO algorithm has better performance.

**Weaknesses:**

1. If the weighted averaged model has better performance than initial model, it is obvious that the aligned model by the PPO algorithm has better performance than others.

**Questions:**

1. It is well known that the neural network is non-convex, which means that applying weighted averaging on several models could lead to a poor performance. Is there always good performance when applying model soup for the reference model? Or how do we avoid this when merging models?

---

> ### Author Response · Authors · 2024-11-22
> **Response to reviwer**
>
> Thanks for your thoughtful review, here are our answers to your concerns. Please let us know if you need any further discussions.
>
> >If the weighted averaged model has better performance than initial model, it is obvious that the aligned model by the PPO algorithm has better performance than others.
>
> Thanks for bringing this to our attention. The same fact that the initial model resides in a better reward model is found within our paper, and we have discussed it in length in section 4.2 Reward Analysis. We first showed in figure 2.a, 2.b and 2.c that the reward is improving as we get closer to the average of SFT models. This is a new phenomenon, and was kind of unexpected to us as the reward model is something that is trained independent of SFT models. We then showed using this phenomenon reward of PPO will improve. So we think it should actually be considered as a strength.
>
> >It is well known that the neural network is non-convex, which means that applying weighted averaging on several models could lead to a poor performance. Is there always good performance when applying model soup for the reference model? Or how do we avoid this when merging models?
>
> This is a great question. Previous research [1,2] has found out fine-tuned models often lie in a similar loss basin, suggesting interpolating between these models can yield better performance than individuals. [1] has shown empirically and theoretically that ImageNet accuracy of vision transformers improves by model averaging. In our research we further extended this to show reward is always improved when doing averaging. Please check Figure 2.c for example, where the centroid of triangle of 3 SFT model resides in a higher reward space.
>
> [1] Model soups: averaging weights of multiple fine-tuned models improves accuracy without increasing inference time, Wortsman et al
>
> [2] What is being transferred in transfer learning? Neyshabur et al

---

> > ### Author Response · Authors · 2024-12-03
> >
> > We noticed that we haven’t heard back from you regarding our response to your review. If you feel that we have sufficiently addressed your concerns, we would greatly appreciate it if you could consider reflecting this in your score.
> >
> > If you have any additional concerns or suggestions, we would be happy to address them. Thank you again for your valuable feedback and time!

---

### Official Review · Reviewer_zFiC · 2024-11-08

**Soundness:** 2
**Presentation:** 3
**Contribution:** 2
**Rating:** 5
**Confidence:** 2

**Summary:**

The paper studies RLHF and proposes SALSA, which leverages a "model soup" approach, averaging weights from two supervised fine-tuned (SFT) models to create a more flexible reference model. They claim that this allows broader exploration of the solution space without sacrificing stability. In the experiments, the authors use models like Llama2-7B, Mistral-7B, and Gemma-2B, along with benchmarks (MT-Bench, Arena-Hard, UltraFeedback, demonstrating that SALSA consistently outperformsPPO by achieving higher rewards and win rates.

**Strengths:**

The idea of this paper is clear and straightforward. The experiment results also seem promising.

**Weaknesses:**

The authors didn't provide sufficient explanations about the occurred phenomenon (see the Questions part), which makes their method not convincing enough. It would be great if the authors can provide some theoretical analysis, even for a simple case study.

**Questions:**

1. Why does the mixed model work the best at the midpoint? It seems that both SFT models are trained randomly, and it seems strange that always the midpoint of them performs the best.

2. For the KL hack phenomenon, is it useful to fix the length of the generated response of the model?

3. The value of the optimal $\beta$ also changes drastically with $\alpha$. When $\alpha = 0.2$ the optimal $\beta$ is 0.2. However, when $\alpha=0.01$, the optimal $\beta$ decreases to $0.01$ quickly. It is also strange that when you take the midpoint model, the KL divergence term seems unnecessary and the trained model won't produce gibberish responses. Could the authors explain more about this phenomenon?

---

> ### Author Response · Authors · 2024-11-22
> **Response to reviewer**
>
> Thanks for showing interest in our work and giving insightful feedback, these are some clarifications and hope we can discuss any remaining concerns:
> >Why does the mixed model work the best at the midpoint? It seems that both SFT models are trained randomly, and it seems strange that always the midpoint of them performs the best.
>
> Here is the model soups equation: $\pi_{\text{soup}} = \alpha \cdot \pi_0 + (1 - \alpha) \cdot \pi_1$
> * As shown in Figures 2a, 2b, and 2c, our empirical results indicate that the reward achieves its highest value at the midpoint between the SFT models. In Figures 2a and 2b, we specifically compared the reward at $\alpha = 0.5$ (midpoint) with other values of $\alpha$ in the range $\{0, 0.1, 0.2, 0.3, 0.4, 0.6, 0.7, 0.8, 0.9\}$. The results consistently show that $\alpha = 0.5$ yields the highest reward. While the actual optimal value may vary slightly (e.g., $\alpha = 0.48$ or $\alpha = 0.53$) depending on the specific models and datasets, the general trend remains clear: as the parameter approaches the midpoint, the reward improves.
> * Previous research [1,2] has shown that fine-tuned models often reside in a similar loss basin, suggesting that interpolating between these models can lead to better performance than using individual models alone. Specifically, as both points lie within the same loss basin, moving closer to their average improves the loss basin. Additionally, the theoretical analysis in Section 4 of [1] supports this observation, favoring an interpolation with $\alpha = 0.5$.
>
> > For the KL hack phenomenon, is it useful to fix the length of the generated response of the model?
>
> Fixing the length of generation is often not effective, as some responses require shorter outputs while others benefit from longer ones.
>
> > It is also strange that when you take the midpoint model, the KL divergence term seems unnecessary and the trained model won't produce gibberish responses. Could the authors explain more about this phenomenon?
>
> The KL divergence coefficient, $\beta$, determines how far the model can deviate from the reference point. An intuitive way to think about effect of beta is drawing n dimensional spheres around reference points. A higher $\beta$ results in a smaller sphere, while a lower $\beta$ allows for a larger sphere. Any point within the line between SFT1 and SFT2 is a valid reference point. Since model soup is in the middle of this line, having a sphere with radius of distance between model soup and SFT1 will contain lots of good reference points that we need to be close to just one of them. Therefore, we can use a larger sphere around model soup (lower $\beta$) to not diverge from the line between two SFT models, but for one SFT point we have less flexibility since we must be close to that exact one point.
>
> For a very simple case suppose we are searching in 2 dimensional space of Figure 2c.  A circle centered at the triangle's centroid, tangent to its edges, encompasses models that are all valid reference points. In contrast, at the vertices of the triangle (representing each SFT model), a circle with same size would extend beyond the region of valid reference points (the triangle). Searching in that big circle might result getting far from any point within triangle which are our good reference points. Therefore, only a small circle (high $\beta$) can be used to ensure the search remains within a meaningful range.
>
>
> > The value of the optimal beta also changes drastically with alpha ...
>
> The gap between beta for alphas of 0.2 and 0.4 are reported in section 4.4(ablations).  The ablation itself was about how the win rate changes over alphas. And the $\beta$ parameters there are documentation of that experiment, it was not supposed to be a thorough analysis of beta vs alpha. The reason for drastic drop from is two folds:
> 1. The granularity of points we found optimal $\beta$ is large within ablation studies (0.2 distance each), where 0.4 is close to center almost and 0.2 is almost close to SFT0.
> 2. We are looking at the optimal win rate for choosing optimal beta, so among a few options (5 points) and a few betas (5 over our search) such patterns a sharp pattern can happen for maximizing a function (win rate here) and doesn't have statistical significance.
>
> A thorougher analysis with more values can result in a more visible pattern, but as we mentioned earlier the goal of that section was finding general pattern of win rates over alphas.
>
> [1] Model soups: averaging weights of multiple fine-tuned models improves accuracy without increasing inference time, Wortsman et al
>
> [2] What is being transferred in transfer learning? Neyshabur et al

---

> > ### Comment · Reviewer_zFiC · 2024-11-25
> > **almost zero $\beta$ when taking the midpoint mixture**
> >
> > Thank you for your response! However, I am still confused why the beta can be almost zero when you take the midpoint model. Let us consider an extremely simplified case where the parameter of the model is two dimensional. Then suppose your two SFT models are $\theta_1, \theta_2$. In your response you mean that the models along the line $(\theta_1, \theta_2)$ are good? However, even if you take the midpoint model, suppose the trained model after RLHF lies within a circle centered at the midpoint model, then it can still deviate from the line (e.g., when the RLHF direction is orthogonal to the line). Therefore, it seems that the KL divergence is still quite necessary.

---

> > > ### Author Response · Authors · 2024-11-30
> > > **Response**
> > >
> > > Thanks for responding to our clarification, there are two reasons:
> > >
> > > since model soups resides in a better area going orthogonal to that direction is also gonna be better. For example check figure 2.c of our paper where the reward in a circle centered around average of SFT1 and SFT2 is gonna be good in two directions. Also please check figure 2 of original model soups paper (https://arxiv.org/pdf/2203.05482), where it is visible that loss is better in a circle around the averaged point.
> > > There are more possible directions that can go right, so when with lower beta the chances of being in the right direction is more than the original case. (The number of degrees that are close to line in that circle are gonna be more than original case)

---

> > > > ### Comment · Reviewer_zFiC · 2024-12-02
> > > >
> > > > Thank you for your feedback! I indeed want more theoretical explanation of this phenomenon instead of only empirical observations because this is counter-intuitive. Like why the models within the circle centered at the average model all perform well. This is weird to me because your SFT models are random and thus it is possible that your trained SFT model happen to be an average model of other two possible SFT models. In this case, isn't exploration around this SFT model better?

---

> ### Author Response · Authors · 2024-12-04
> **New Response**
>
> Hi,
> As we agree this is counterintuitive the theoretical explanation of why model soups are performing better is considered a previous work and already known in the literature. What we have discovered newly is when training reward models the same phenomenon happens again and we can use it to improve RLHF. The same reasoning will go by replacing loss function to reward model.
>
> Under the model averaging literature [1] averages parameters for improved generalization, they also provides theories on why this is improving functions around averaged models. [2,3] show OOD performance by averaging models finetuned with different hyperparams. [4] shows regions around close models contains potentially superior models. [5] improves base models by merging finetuned models. [6] frequently averages separately trained models.
>
> Under mode connectivity literature, there are also numerous examples. [7] shows finetuned models resides in the same loss basin. Because they are in the same loss basin, the center of them will look like the center of watermelon, and the two models will be like the edges of it. [8] shows model share a loss valley even up to a permutation.
>
> [1] Averaging Weights Leads to Wider Optima and Better Generalization https://arxiv.org/pdf/1803.05407
>
> [2] Model soups: averaging weights of multiple fine-tuned models improves accuracy without increasing inference time https://arxiv.org/pdf/2203.05482
>
> [3] Diverse Weight Averaging for Out-of-Distribution Generalization https://arxiv.org/abs/2205.09739
>
> [4] Knowledge is a Region in Weight Space for Finetuned Language Models https://arxiv.org/pdf/2302.04863
>
> [5]  Fusing finetuned models for better pretraining https://arxiv.org/abs/2204.03044
>
> [6] PopulAtion Parameter Averaging (PAPA) https://arxiv.org/pdf/2304.03094
>
> [7] What is being transferred in transfer learning? https://arxiv.org/abs/2008.11687
>
> [8] Random initialisations performing above chance and how to find them https://arxiv.org/pdf/2209.07509

---

### Author Response · Authors · 2024-11-22
**Updated paper**

We have updated the paper including the following changes:
* Added KL training curves to appendix per reviewers request.
* Improved quality of figures
* Rewrote the main results section for better understandability
* Moved the KL divergence discussion to Appendix

---

### Meta-Review · Area_Chair_F9GY · 2024-12-21

**Metareview:**

This paper proposes a model-soup technique which uses the average weight to construct the KL center in RLHF. The majority of the reviewers vote for rejection. The main complains are missing sufficient, rigorous explanation of the phenomena and the current empirical finding has been shown only for PPO  and on the selected dataset. The authors provide empirical analyses and argue that the same can be applied to DPO, though the answers are not satisfactory. Some reviewers also raise similarity between this work and other weight-averaging approach to RLHF, though I agree with the authors the algorithms are different.

Given the reviews, I decide to not recommend publication. I think the idea here is quite interesting and potentially points out big phenomenon. But I agree with the reviewers that more evidences are needed theoretically or empirically (more datasets, more algorithms).

**Additional Comments On Reviewer Discussion:**

Reviewer zFiC, Reviewer pSBg, Reviewer T2cX raise concerns on missing sufficient explanation; the authors provide some empirical evidence and cite related work of model averaging but lacks a theoretical or rigorous explanation on the working principle. Reviewer T2cX also raise issues about missing important related work on model soups with RLHF; the authors discuss the differences.

Reviewer NSBF, Reviewer K7rw, and Reviewer yT8z raise question of experiments limited to PPO, asking if the same applies to DPO e.g. The authors mention PPO is found to superior than DPO, but this answer is not satisfactory. In addition, Reviewer K7rw raises clarity questions, and they are addressed. Reviewer NSBF complain the lack of experiments, details, and comparison with other related work. The authors's response address only partially; the new fig 9 does not provide substantial difference in reward.

---

### Decision · Program_Chairs · 2025-01-22

Reject